# SaLon3R: Structure-aware Long-term Generalizable 3D Reconstruction from Unposed Images

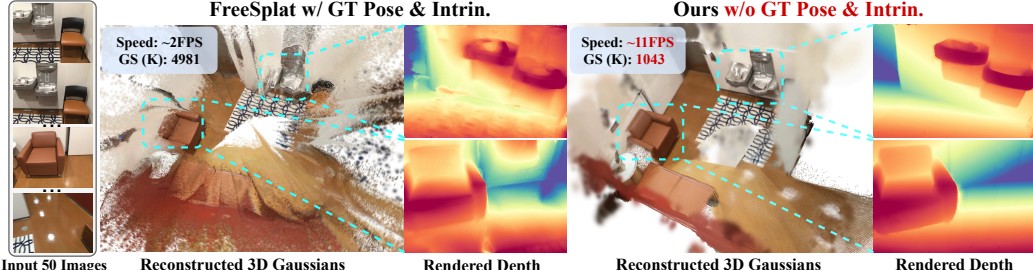

Figure 1: **Comparison between FreeSplat (Wang et al., 2024c) and our SaLon3R.** Given a sequence of *unposed and uncalibrated* images as input, our method achieves online generalizable Gaussian reconstruction with over 10 **FPS speed**, allowing high-quality rendering, accurate depth estimation, effectively reducing GS from 9830K to 1043K with nearly **90**% **redundancy removal.**

## Abstract

Recent advances in 3D Gaussian Splatting (3DGS) have enabled generalizable, on-the-fly reconstruction of sequential input views. However, existing methods often predict per-pixel Gaussians and combine Gaussians from all views as the scene representation, leading to substantial redundancies and geometric inconsistencies in long-duration video sequences. To address this, we propose **SaLon3R**, a novel framework for **S**tructure-**a**ware, **Lon**g-term 3DGS **R**econstruction. To our best knowledge, SaLon3R is the first online generalizable GS method capable of reconstructing over **50** views in over **10** FPS, with **50**% **to 90**% redundancy removal. Our method introduces *compact anchor primitives* to eliminate redundancy through differentiable *saliency-aware Gaussian quantization*, coupled with a *3D Point Transformer* that refines anchor attributes and saliency to resolve cross-frame geometric and photometric inconsistencies. Specifically, we first leverage a 3D reconstruction backbone to predict dense per-pixel Gaussians and a saliency map encoding regional geometric complexity. Redundant Gaussians are compressed into compact anchors by prioritizing high-complexity regions. The 3D Point Transformer then learns spatial structural priors in 3D space from training data to refine anchor attributes and saliency, enabling regionally adaptive Gaussian decoding for geometric fidelity. Without known camera parameters or test-time optimization, our approach effectively resolves artifacts and prunes the redundant 3DGS in a single feed-forward pass. Experiments on multiple datasets demonstrate our state-of-the-art performance on both novel view synthesis and depth estimation, demonstrating superior efficiency, robustness, and generalization ability for long-term generalizable 3D reconstruction. Code will be released.

## 1 Introduction

In recent years, Neural Radiance Fields (NeRF) (B. Mildenhall *et al.*, 2021) and 3D Gaussian Splatting (3DGS) (Kerbl et al., 2023) have significantly advanced novel view synthesis (NVS), which

aims to reconstruct the 3D scene from multi-view images and render high-fidelity images when queried with novel views. However, these methods are limited to per-scene test time optimization and lack the generalization ability to unseen data. To address this, generalizable models (Yu et al., 2021; A. Chen *et al.*, 2021; Guo et al., 2024; Chen et al., 2025) have been proposed to enable a feed-forward reconstruction, achieving generalizable view synthesis. These methods require an additional pre-processing stage to estimate camera poses, *e.g.*, Structure-from-Motion. Besides its large time consumption, this pre-processing stage is sensitive to sparse and in-the-wild views, leading to inaccurate estimation and degraded rendering. To get rid of this reliance, some pose-free generalizable approaches (Smith et al., 2023; Chen & Lee, 2023; Li et al., 2024; Hong et al., 2024) are proposed to realize both pose estimation and reconstructing the 3D scene.

Despite their promising results, the major drawback of the current generalizable methods is the lack of multi-view Gaussian fusion with redundancy removal, limiting their applications to long-term and large-scale scene reconstruction. Specifically, given the context images as input, most methods typically predict per-pixel Gaussian attributes and combine Gaussians from all views as the scene representation, without additional structure-aware inductive bias to refine the redundancy or artifacts. It results that previous methods either fail due to large GPU memory consumption by a large number of redundant Gaussians or experience a degradation in rendering quality due to photometric inconsistency and 3D misalignment arising from reconstruction inaccuracies when addressing long-duration video images. To address this issue, some recent works (Wang et al., 2024c; Fei et al., 2024; Ziwen et al., 2024) present additional adaptation to reduce the Gaussian redundancy by fusing the overlapping 3D Gaussians in image space. However, they mainly utilize 2D features for aggregating the overlapped regions, and lack a holistic structure-aware 3D understanding of the scene to control the adaptive distribution of the 3D Gaussians. This will inevitably limit their application to longer-term (*i.e.*, over 50 frames) generalizable Gaussian reconstruction, yielding low efficiency and a large memory cost due to the accumulation of pixel-wise Gaussians, as shown in Fig. 1.

In this paper, we present **SaLon3R** for **S**tructure-**a**ware, **Lon**g-term **3**DGS **R**econstruction from *unposed* and *uncalibrated* images, achieving online scene representation learning in a feed-forward manner. To our best knowledge, SaLon3R is the first online generalizable GS method capable of reconstructing over 50 views at a speed exceeding 10 FPS. The core of our method is to encode the pixel-wise Gaussians to compact anchor primitives, leveraging spatial structural priors to eliminate redundancy and refine artifacts in a single feed-forward pass. Specifically, we adopt CUT3R (Wang et al., 2025b) as the backbone to predict global pointmaps, per-pixel Gaussian latents, and associated saliency map. Unlike previous methods that fuse the overlapped regions, we devise a saliency-aware quantization mechanism to encode the pixel-wise dense Gaussians into compact Gaussian anchors by voxel aggregation. To address the inconsistencies, we utilize a lightweight point transformer that applies local attention to serialized anchors, effectively capturing 3D spatial relationships to resolve artifacts and inconsistencies. After the refinement, the Gaussian anchors are adaptively decoded into Gaussians based on the updated 3D saliency of each voxel, allowing adaptive Gaussian density control. Our model enables the extrapolation rendering ability for improved quality and robustness.

Extensive experiments show that our method significantly outperforms pose-free approaches and surpasses pose-required methods in depth estimation, while achieving competitive performance in novel view synthesis. It further achieves 50% to 90% redundancy reduction and online reconstruction at more than 10 FPS. Moreover, our method exhibits strong generalization, outperforming pose-required methods in zero-shot settings. The contributions of this work are summarized as follows.

- We introduce SaLon3R for pose-free long-term generalizable reconstruction with structure-aware learning to eliminate redundancy and refine artifacts, enhancing the rendering quality and geometric consistency. To our best knowledge, SaLon3R is the first online generalizable GS method capable of reconstructing over 50 views at a speed exceeding 10 FPS.

- We present a saliency-aware Gaussian quantization mechanism to encode the dense pixel-wise Gaussians into sparse Gaussian anchors incrementally. A lightweight point transformer is employed to achieve interaction between the surrounding Gaussians to learn the 3D structure, allowing out-of-distribution prediction and artifacts refinement.

- Extensive experiments demonstrate our superior performance on long-term generalizable online reconstruction from unposed and uncalibrated images. With 50% to 90% redundancy removal, our SaLon3R significantly outperforms pose-required methods in depth estimation and achieves state-of-the-art results in novel view synthesis.

## 2 RELATED WORK

**Generalizable Novel View Synthesis.** NeRF (B. Mildenhall *et al.*, 2021) and 3DGS (Kerbl et al., 2023) have revolutionized the field of novel view synthesis and 3D reconstruction. Recent methods empower 3DGS with feed-forward prediction to achieve generalizable view synthesis (Charatan et al., 2024; Chen et al., 2025; Smart et al., 2024; Ye et al., 2024; Zhang et al., 2025). However, most existing methods focus on sparse view reconstruction, combining pixel-wise Gaussian representations from all context views as the global representation. However, 3DGS from different views cannot be directly merged due to the view-dependent rendering process, which may bring photometric and geometric inconsistencies and lead to performance degeneration. To address this, FreeSplat (Wang et al., 2024c) introduces cross-view feature aggregation and pixel-wise triplet fusion to eliminate redundancy. PixelGaussian (Fei et al., 2024) dynamically adjusts both the distribution and the number of Gaussians based on the geometric complexity. Long-LRM (Ziwen et al., 2024) adopts an efficient token-based strategy combined with Gaussian pruning to manage long sequential data. However, these methods mainly focus on fusing the overlapping regions of the pixel-wise Gaussians among views, which lack a structure-aware 3D understanding of the scene to adaptively control the global distribution of 3DGS. In contrast, our method can effectively reduce the redundancy and mitigate the geometrical and photometric inconsistencies, offering a more promising, scalable, and efficient solution to handle unposed long-term images.

**3D Reconstruction Network.** Recent advances in learning-based 3D reconstruction pave the way for the joint estimation of 3D geometry and camera parameters in an end-to-end framework (Wang et al., 2024b; Leroy et al., 2024; Wang et al., 2025b;a; Guo et al., 2025). This could avoid the accumulated errors across decomposed multiple tasks in traditional reconstruction pipelines, such as feature extraction (DeTone et al., 2018; Yi et al., 2016), feature matching (Lindenberger et al., 2023; Sarlin et al., 2020), and SfM (J. Schonberger *et al.*, 2016; Wang et al., 2024a). To address these limitations, DUSt3R (Wang et al., 2024b) proposes a unified framework that reformulates stereo reconstruction as a pointmap regression task, eliminating the reliance on explicit camera calibration. MASt3R (Leroy et al., 2024) further enhances the matching accuracy by introducing a dense feature prediction head trained with an additional matching supervision loss. CUT3R (Wang et al., 2025b) leverages latent states to generate metric point maps for each incoming image in an online manner. VGGT (Wang et al., 2025a) proposes a feed-forward neural network to directly estimate the key 3D attributes from hundreds of views. To achieve incremental Gaussian reconstruction, we adopt CUT3R (Wang et al., 2025b) and VGGT (Wang et al., 2025a) as reconstruction backbones to achieve both online and offline novel view synthesis from unposed, long-sequence image inputs.

**3DGS with Quantization.** To address the large amount of 3DGS, some quantization-based methods aim to compress the 3DGS representation for efficiency while preserving visual quality. CompGS (Liu et al., 2024) introduces a hybrid primitive structure to improve compactness with a compression ratio up to 110x. Similarly, LightGaussian (Fan et al., 2024) combines vector quantization with knowledge distillation and pseudo-view augmentation, reducing model size. Scaffold-GS (Lu et al., 2024) introduces an anchor-based approach, voxelizing scenes to create representative anchors that derive Gaussian attributes. Octree-GS (Ren et al., 2024) utilizes an octree-based structure to hierarchically partition 3D space. In our work, we introduce a saliency-aware Gaussian quantization mechanism, achieving $50\%$ to $90\%$ redundancy removal at a speed exceeding 10 FPS.

## 3 METHODOLOGY

### 3.1 OVERVIEW

Given a stream of $N$ *unposed* and *uncalibrated* images as input, our SaLon3R aims to learn a feed-forward network to enable online reconstruction for 3D Gaussian Splatting $\mathcal{G} = \{\boldsymbol{\mu}, \boldsymbol{\Sigma}, \boldsymbol{\alpha}, \mathbf{sh}\}$ with mean $\boldsymbol{\mu}$, covariance $\boldsymbol{\Sigma}$, opacity $\boldsymbol{\alpha}$, and spherical harmonics coefficients $\mathbf{sh}$. Mathematically, we aim to learn the following online reconstruction:

$$\mathcal{G}_t, \{\mathbf{K}_t, \mathbf{P}_t\}_{t=1}^N = \text{SaLon3R}(\mathbf{I}_t, \mathcal{G}_{t-1}), \tag{1}$$

where $\mathbf{K}_t$ and $\mathbf{P}_t$ indicate the estimated camera intrinsic and extrinsic. As shown in Fig. 2, given every single frame as input, the online reconstruction network predicts the corresponding pointmap in the world coordinate, per-pixel Gaussian latent, and saliency map (Sec. 3.2). To eliminate the

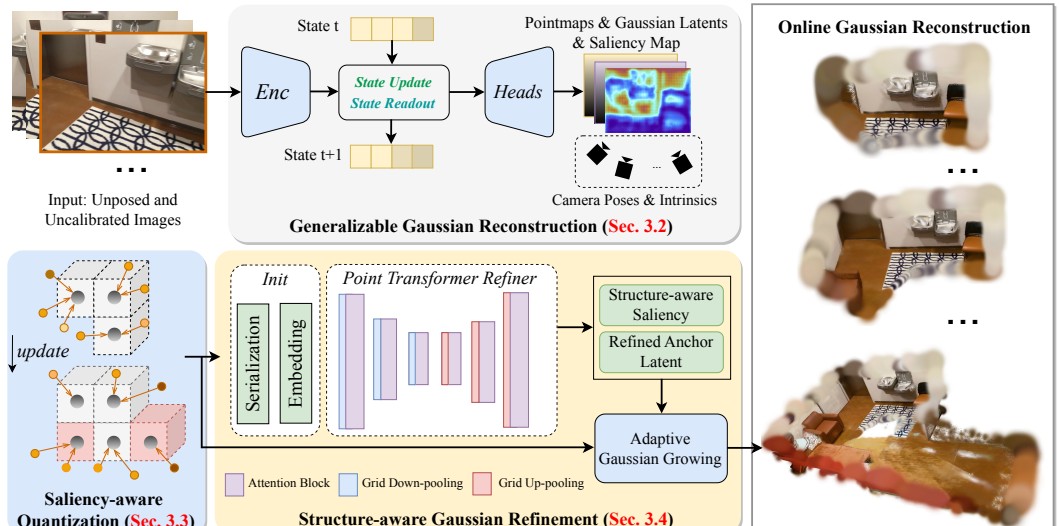

Figure 2: **Overview of our proposed SaLon3R.** Given every unposed and uncalibrated image as input, an incremental reconstruction network initially predicts the camera parameters $\mathbf{K}_t, \mathbf{P}_t$, pointmaps $\hat{\mathbf{X}}_t$, and pixel-wise Gaussian latents $\hat{\mathbf{G}}_t$. A saliency-aware quantization mechanism is designed to encode dense Gaussians into compact Gaussian anchors, reducing redundancy while maintaining geometric fidelity. A lightweight point transformer is adopted to enable structure-aware Gaussian refinement by capturing the spatial relationships to resolve the inconsistencies and enhance the extrapolation ability, achieving adaptive Gaussian growing with enhanced rendering quality.

redundancy of pixel-wise Gaussians, we devise a saliency-aware quantization mechanism to encode the pixel-wise dense Gaussians into compact Gaussian anchors by voxel downsampling (Sec. 3.3). To address the potential artifacts in reconstruction, we employ a lightweight point transformer to enable local attention of the serialized anchors to capture spatial relationships between anchors, allowing a structure-aware learning of the Gaussian field. Finally, we decode the anchors to Gaussians with adaptive density control, using the refined structure-aware saliency from the refiner (Sec . 3.4). To achieve online reconstruction, during inference, only part of the global Gaussian $\mathcal{G}_{t-1}$ that are near the frustum of the current view are extracted to be merged with the current pixel-wise Gaussian latents as input for quantization and refinement to incrementally update to $\mathcal{G}_t$.

## 3.2 GENERALIZABLE GAUSSIAN RECONSTRUCTION

**Preliminary of CUT3R (Wang et al., 2025b).** Given a sequence of streaming unposed and uncalibrated images $\{\mathbf{I}_t\}_{t=1}^N$, we first encode the image to tokens $\mathbf{F}_t$ via ViT encoder. For incremental reconstruction, the state is first initialized as learnable tokens to interact with image tokens by state-update and state-readout. Specifically, state-update utilizes input image tokens to update the state $\mathbf{s}_{t-1}$ to $\mathbf{s}_t$, and state-readout aims to read the context from the state to provide historical spatial information. The two interactions are conducted with two interconnected transformer decoders:

$$[\mathbf{z}'_t, \mathbf{F}'_t], \mathbf{s}_t = \text{Decoders}([\mathbf{z}_t, \mathbf{F}_t], \mathbf{s}_{t-1}), \tag{2}$$

where $\mathbf{z}$ refers to the pose token to learn motion information for pose estimation. Both image token $\mathbf{F}_t$ and pose token $\mathbf{z}_t$ are updated to $\mathbf{F}'_t$ and $\mathbf{z}'_t$ with updated context information.

**Geometry and 3DGS Decoding.** Subsequently, geometric results and camera poses are estimated from $\mathbf{z}'_t$ and $\mathbf{F}'_t$ to predict pointmaps and confidence maps. Similarly, we adopt the DPT-like architecture (Ranftl et al., 2021) as Gaussian head to predict the pixel-wise Gaussian latents $\hat{\mathbf{G}}_t \in \mathbb{R}^{H \times W \times C}$ with associated saliency map $\mathbf{S}_t$, which captures the geometric and photometric complexity of different regions spatially. Specifically, the predicted local pointmaps $\hat{\mathbf{X}}_t^{\text{self}}$ and camera poses $\hat{\mathbf{P}}_t$ are utilized to compute the pointmaps in world coordinate as Gaussian centers $\hat{\mathbf{X}}_t \in \mathbb{R}^{H \times W \times 3}$. Besides, we estimate focal length with the Weiszfeld algorithm (Plastria, 2011).

$$\hat{\mathbf{X}}_t^{\text{self}} = \text{Head}_{\text{pts}}(\mathbf{F}'_t), \quad \hat{\mathbf{G}}_t = \text{Head}_{\text{GS}}(\mathbf{F}'_t), \quad \hat{\mathbf{P}}_t = \text{Head}_{\text{pose}}(\mathbf{z}'_t). \tag{3}$$

### 3.3 SALIENCY-AWARE GAUSSIAN QUANTIZATION

Despite the advances of current generalizable methods, they are limited to the sparse view prediction due to directly merging the pixel-wise Gaussians with view-dependent effects, leading to photometric and geometric inconsistencies, slow inference speed, and large GPU allocation. While recent methods (Wang et al., 2024c; Fei et al., 2024) mitigate the problem by fusing the overlapped region, they fall short of a global 3D structure understanding to reduce redundancy while preserving geometric fidelity. To address the issue, we devise a saliency-aware quantization for points aggregation, allowing a significant redundancy removal while avoiding the information loss in salient regions, *i.e.*, the region with rich textures and geometry complexity.

**Voxelization for Gaussian Centers.** Given every image as input, we build the anchors with the predicted Gaussian latents $\hat{\mathbf{G}}_t$ and Gaussian centers $\hat{\mathbf{X}}_t$. Specifically, we first voxelize the scene from the Gaussian centers $\hat{\mathbf{X}}_t$ and compute the voxel center $\mathbf{v}_i$ for each point $\mathbf{x}_i \in \hat{\mathbf{X}}_t$: $\mathbf{v}_i = \left\lfloor \frac{\mathbf{x}_i}{\gamma} \right\rfloor \cdot \gamma$, where $\lfloor \cdot \rfloor$ denotes the floor operation, and $\gamma$ is the voxel size for grid resolution. From this operation, we effectively reduce the redundancy by quantizing the pointmap to a set of voxels.

**Saliency-based Anchor Fusion.** After voxelization, we aggregate the points within the same voxel to a single anchor representing the geometry and the Gaussian latent. Each point $\mathbf{v}_i$ is associated with a predicted global saliency score $s_i$ to indicate its appearance and geometry complexity. It is noted that the saliency estimation is learnable and only trained with rendering loss without any ground truth. We use the saliency scores to compute a normalized weighted sum to fuse the anchors for positions and latents, ensuring more salient points contribute more significantly to the fused representation. We take the fused anchor position as the anchor Gaussian center $\boldsymbol{\mu}_k^a$. For a voxel with $L$ points, we fuse the anchor feature as follows:

$$\mathbf{s}_k^a = \sum_{i=1}^{L} s_i, \quad \boldsymbol{\mu}_k^a = \frac{\sum_{i=1}^{L} s_i \mathbf{v}_i}{\mathbf{s}_k^a}, \quad \mathbf{f}_k^a = \frac{\sum_{i=1}^{L} s_i \mathbf{f}_i}{\mathbf{s}_k^a}. \tag{4}$$

After the anchor fusion, a MLP head is employed to convert the anchor latent $\mathbf{f}_k^a$ to anchor Gaussian attributes as: $\{\boldsymbol{\Sigma}_k^a, \boldsymbol{\alpha}_k^a, \mathbf{sh}_k^a\} = \text{MLP}_{\text{GS}}(\mathbf{f}_k^a)$.

### 3.4 STRUCTURE-AWARE GAUSSIAN REFINEMENT

Given multiple views as input, directly fusing predicted Gaussians into a coherent scene representation is insufficient to address the photometric inconsistencies across views and 3D misalignments arising from reconstruction and pose inaccuracies. Additionally, while current generalizable methods support effective interpolation to render novel views within the context views, it is also crucial to enable the model to learn extrapolation prediction to enhance the robustness of extrapolated camera views. To address this, we adopt Point Transformer V3 (PTV3) (Wu et al., 2024) as an efficient structure-aware point cloud encoder to capture spatial relationships in predicted coarse anchor Gaussians, eliminating the artifacts while enhancing the 3D consistency. By training on large-scale datasets with out-of-distribution views, our model also enhances the extrapolation rendering ability for better rendering quality and robustness.

**Point Cloud Serialization.** To enable efficient pointcloud processing, we first use pointcloud serialization to transform the unstructured point-based anchors to a structured format. Specifically, we employ the serialized encoding to convert the point position to an integer reflecting the order in a space-filling curve (*i.e.*, z-order curve and Hilbert curve). After serialization, the input points are sparsified and encoded to features through an embedding layer.

**Point Transformer Refiner.** The point transformer follows the structure of U-Net, consisting of a five-stage encoder with a downsampling grid pooling layer and attention blocks in each stage, and a decoder with an upsampling grid pooling layer and attention blocks in each stage. For each attention block, we apply patch attention to group points into non-overlapping patches, performing interaction among patches for attention computation. Specifically, we concatenate the Gaussian attributes with the anchor saliency as the feature to the input, and the refined anchor latent is predicted as:

$$\mathbf{h}_k^a = \text{PointTransformer}(\{\boldsymbol{\mu}_k^a, \boldsymbol{\Sigma}_k^a, \boldsymbol{\alpha}_k^a, \mathbf{sh}_k^a, \mathbf{s}_k^a\}). \tag{5}$$

**Adaptive Gaussian Growing.** To derive the Gaussians from anchor latents, we apply a growing strategy to predict $M$ Gaussians from each anchor. The residual primitives are decoded through an

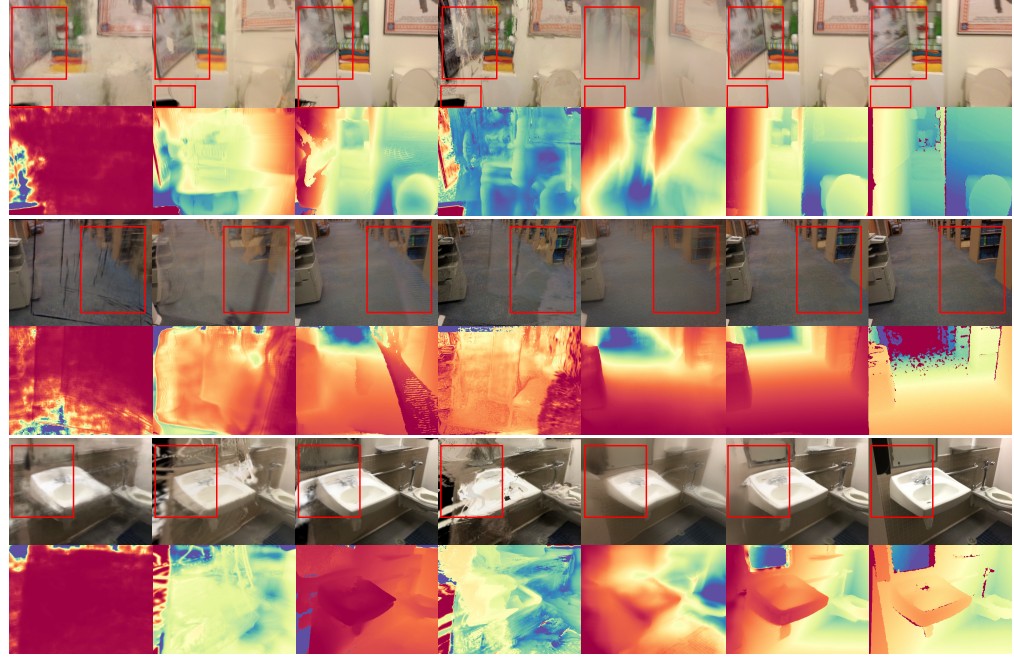

| PixelSplat | MVSplat | FreeSplat | PixelGaussian | NoPoSplat | **Ours** | **GroundTruth** |

Figure 3: **Qualitative Comparison on ScanNet dataset.** Given 10 context views as input, we compare the novel view synthesis results for both rendered color and depth. We also provide more qualitative results in the supplementary material.

MLP head from the refined anchor latent $\mathbf{h}_k^a$:

$$\{\Delta\boldsymbol{\mu}_{k,i}, \Delta\boldsymbol{\Sigma}_{k,i}, \Delta\boldsymbol{\alpha}_{k,i}, \Delta\mathbf{sh}_{k,i}, \Delta\mathbf{s}_{k,i}\}_{i=1}^M = \text{MLP}_{\text{grow}}(\mathbf{h}_k^a). \tag{6}$$

To achieve adaptive density control, our goal is to prune redundant Gaussians in regions with sparse textures and low geometric complexity, while allocating more Gaussians to recover high-frequency details and regions with large depth gradients. To this end, we leverage the refined saliency, which encodes structure-aware spatial information through the point transformer. Specifically, the saliency is fused to update the opacity, and a mask $\mathbf{M}^r$ is derived to filter out redundant Gaussians.

$$\boldsymbol{\alpha}_{k,i}^r = (\boldsymbol{\alpha}_{k,i}^a + \Delta\boldsymbol{\alpha}_{k,i}) \cdot (1 + \text{Tanh}(\mathbf{s}_{k,i}^a + \Delta\mathbf{s}_{k,i})), \quad \mathbf{M}_{k,i}^r = \boldsymbol{\alpha}_{k,i}^r > \beta, \tag{7}$$

where $\beta$ is the threshold for opacity pruning. We could obtain the refined Gaussian prediction $\mathcal{G}^r$:

$$\mathcal{G}^r = \bigcup_{k=1}^K \left\{ \left(\boldsymbol{\mu}_k^a + \Delta\boldsymbol{\mu}_{k,i}, \boldsymbol{\Sigma}_k^a + \Delta\boldsymbol{\Sigma}_{k,i}, \boldsymbol{\alpha}_{k,i}^r, \mathbf{sh}_k^a + \Delta\mathbf{sh}_{k,i}\right) \mid \mathbf{M}_{k,i}^r = 1, i = 1, \ldots, M \right\}. \tag{8}$$

**Training Objective.** We fix the pretrained weights of the reconstruction network (except for Gaussian/saliency prediction) and train other modules. During training, we take randomly sampled views as context views and the others as target views from a sequence, to train our model for handling extrapolation views. We follow (Charatan et al., 2024) to use the perceptual loss and L1 loss as the photometric loss between the rendered target image and ground-truth images. Additionally, to refine the rendered depth smoothness, we apply an edge-aware depth regularization loss on the rendered depth to constrain the surface smoothness. The total loss is defined as:

$$\mathcal{L}_{total} = \lambda_1 \mathcal{L}_1 + \lambda_2 \mathcal{L}_{\text{LPIPS}} + \lambda_3 \mathcal{L}_{\text{smooth}}, \tag{9}$$

where $\lambda_1, \lambda_2, \lambda_3$ denote the weights for different losses, we set $\lambda_1 = 1.0, \lambda_2 = 0.05, \lambda_3 = 0.0005$.

## 4 EXPERIMENTS

### 4.1 EXPERIMENTAL SETUP

**Datasets.** We train our method on the ScanNet (Dai et al., 2017) dataset, following FreeSplat (Wang et al., 2024c) to have 100 scenes for training and 8 scenes for testing. Unlike FreeSplat, we adopt a

Table 1: **Novel View Evaluation results on ScanNet (Dai et al., 2017).** PF: Pose-free methods.

| PF | Method | 10 views | | | 30 views | | | 50 views | | |
|---|---|---|---|---|---|---|---|---|---|---|
| | | PSNR↑ | SSIM↑ | LPIPS↓ | PSNR↑ | SSIM↑ | LPIPS↓ | PSNR↑ | SSIM↑ | LPIPS↓ |
| ✗ | PixelSplat | 17.77 | 0.572 | 0.505 | - | - | - | - | - | - |
| | MVSplat | 17.74 | 0.643 | 0.411 | - | - | - | - | - | - |
| | FreeSplat | 23.24 | 0.785 | 0.266 | 21.16 | 0.704 | 0.342 | 19.90 | 0.702 | 0.365 |
| | PixelGaussian | 17.41 | 0.601 | 0.429 | - | - | - | - | - | - |
| ✓ | NoPoSplat | 18.46 | 0.652 | 0.396 | 17.03 | 0.637 | 0.506 | 15.71 | 0.574 | 0.515 |
| | FLARE | 18.24 | 0.691 | 0.403 | 16.21 | 0.602 | 0.434 | 15.92 | 0.566 | 0.577 |
| | **Ours** | 22.54 | 0.750 | 0.337 | 20.47 | 0.679 | 0.384 | 19.78 | 0.682 | 0.392 |
| | **Ours**$_{\text{w/ VGGT}}$ | 23.26 | 0.805 | 0.289 | 21.25 | 0.728 | 0.384 | 20.26 | 0.720 | 0.377 |

Table 2: **Novel view depth evaluation on ScanNet (Dai et al., 2017).** PF: Pose-free methods.

| PF | Method | 10 views | | | | 30 views | | | | 50 views | | | |
|---|---|---|---|---|---|---|---|---|---|---|---|---|---|
| | | Abs Rel↓ | $\delta_1$ ↑ | FPS↑ | GS↓ | Abs Rel↓ | $\delta_1$ ↑ | FPS↑ | GS↓ | Abs Rel↓ | $\delta_1$ ↑ | FPS↑ | GS↓ |
| ✗ | PixelSplat | 0.680 | 0.715 | 1.15 | 5898 | - | - | - | - | - | - | - | - |
| | MVSplat | 0.331 | 0.811 | 2.11 | 1966 | - | - | - | - | - | - | - | - |
| | FreeSplat | 0.102 | 0.973 | 3.66 | 1082 | 0.411 | 0.894 | 2.87 | 2934 | 0.346 | 0.904 | 2.36 | 4981 |
| | PixelGaussian | 0.392 | 0.766 | 3.05 | 2952 | - | - | - | - | - | - | - | - |
| ✓ | NoPoSplat | 0.390 | 0.794 | 3.59 | 1180 | 1.185 | 0.559 | 1.24 | 3801 | 1.716 | 0.463 | 0.91 | 6422 |
| | FLARE | 0.487 | 0.802 | 3.32 | 1180 | 1.212 | 0.543 | 1.13 | 3801 | 1.814 | 0.489 | 0.85 | 6422 |
| | **Ours** | 0.030 | 0.977 | 10.16 | 866 | 0.060 | 0.953 | 10.10 | 2073 | 0.068 | 0.972 | 10.15 | 3042 |
| | **Ours**$_{\text{w/ VGGT}}$ | 0.030 | 0.978 | 10.53 | 853 | 0.051 | 0.960 | 10.72 | 1972 | 0.062 | 0.974 | 10.66 | 2897 |

more challenging setting by using sparsely overlapping input images, where the context view overlap is limited to only 15%–50%. To validate the generalization ability, we further conduct the cross-dataset evaluation on ScanNet++ (Yeshwanth et al., 2023) and Replica (Straub et al., 2019). For fair comparison, we retrain all the methods on ScanNet (Dai et al., 2017) based on their pretrained weights and adopt the same validation setting as our method.

**Evaluation Metrics.** We evaluate the performance of novel view synthesis with PSNR, SSIM (Z. Wang et al., 2004), and LPIPS (R. Zhang et al., 2018). To evaluate the depth, we report two common metrics in mono-depth estimation: Abs Rel and $\delta < 1.25$ ($\delta_1$). The unit for the 3DGS number is K.

**Baselines.** To validate our performance, we extensively compare SaLon3R with previous feed-forward methods: 1) Pose-required: PixelSplat (Charatan et al., 2024), MVSplat (Chen et al., 2025), FreeSplat (Wang et al., 2024c), PixelGaussian (Fei et al., 2024), and 2) Pose-free: FLARE (Zhang et al., 2025), NoPoSplat (Ye et al., 2024). Due to the out-of-memory problem from large 3DGS occupation, PixelSplat (Charatan et al., 2024), MVSplat (Chen et al., 2025), and PixelGaussian (Fei et al., 2024) fail to report the results on 30 views and 50 views.

**Implementation Details.** All experiments are implemented using Pytorch (Paszke et al., 2019) on A100 NVIDIA GPU. We use the pretrained weights from CUT3R (Wang et al., 2025b) and fix the scene reconstruction module during training. We employ the Adam optimizer (Kingma & Ba, 2014) to train the point transformer, Head$_{GS}$, MLP$_{GS}$, and MLP$_{grow}$. We follow (Wang et al., 2024c) to conduct experiments under resolution as $512 \times 384$ for the ScanNet (Dai et al., 2017) and ScanNet++ (Yeshwanth et al., 2023) datasets. We adopt $M = 4$ for Gaussian growing.

## 4.2 MAIN RESULTS

**Novel View Synthesis.** We evaluate novel view synthesis using 10, 30, and 50 input views, with results reported in Tab. 1 and detailed analysis in the appendix. Our method achieves competitive performance with pose-required approaches and consistently outperforms pose-free methods. As shown in Fig. 3, PixelSplat (Charatan et al., 2024) and MVSplat (Chen et al., 2025) suffer from geometric and photometric inconsistencies, producing blurry RGB and inaccurate depth due to the absence of Gaussian pruning. FreeSplat (Wang et al., 2024c) and PixelGaussian (Fei et al., 2024) mitigate redundancy but only prune in overlapping regions, limiting structure-aware refinement and generalization to extrapolated views. In contrast, our SaLon3R leverages structural priors from

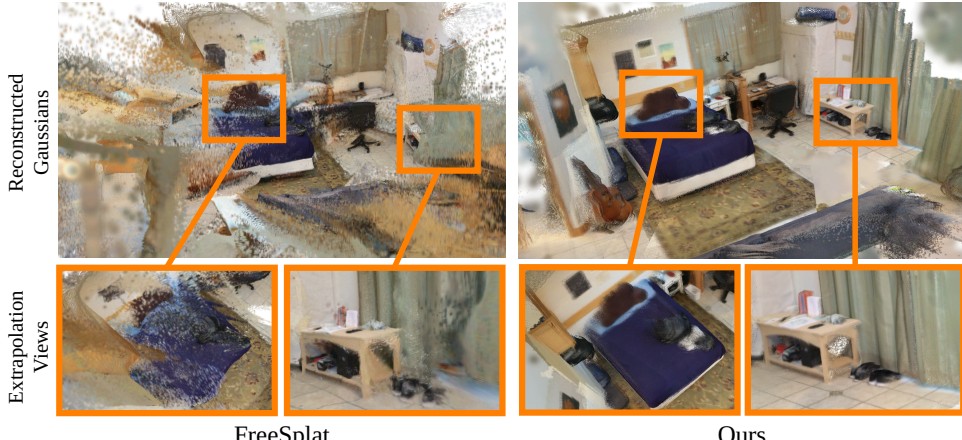

Figure 4: **Visualization of Global Gaussian Splatting and Extrapolation Views.** We zoom in extrapolated views of the bed and desk for better comparison. It demonstrates our method outperforming the FreeSplat with more consistent 3D structure, resulting better rendering performance.

Table 3: **Zero-shot Novel View Synthesis on Replica (Straub et al., 2019) and ScanNet++ (Yeshwanth et al., 2023) from 10 views.**

| PF | Method | Replica | | | | | ScanNet++ | | | | |
|---|---|---|---|---|---|---|---|---|---|---|---|
| | | PSNR↑ | SSIM↑ | LPIPS↓ | Abs Rel ↓ | $\delta_1$ ↑ | PSNR↑ | SSIM↑ | LPIPS↓ | Abs Rel ↓ | $\delta_1$ ↑ |
| ✗ | PixelSplat | 16.61 | 0.587 | 0.607 | 4.248 | 0.343 | 17.54 | 0.625 | 0.523 | 0.766 | 0.530 |
| | MVSplat | 16.85 | 0.642 | 0.436 | 2.119 | 0.610 | 17.01 | 0.606 | 0.446 | 0.875 | 0.577 |
| | PixelGaussian | 16.12 | 0.593 | 0.668 | 2.865 | 0.591 | 17.46 | 0.608 | 0.491 | 1.182 | 0.574 |
| | FreeSplat | 18.92 | 0.719 | 0.369 | 1.485 | 0.790 | 22.56 | 0.769 | 0.258 | 0.350 | 0.828 |
| ✓ | NoPoSplat | 16.72 | 0.674 | 0.446 | 1.723 | 0.792 | 17.02 | 0.667 | 0.546 | 0.658 | 0.647 |
| | FLARE | 16.75 | 0.667 | 0.431 | 1.497 | 0.710 | 16.91 | 0.659 | 0.471 | 0.696 | 0.609 |
| | **Ours** | 20.79 | 0.683 | 0.312 | 0.712 | 0.917 | 22.08 | 0.694 | 0.372 | 0.104 | 0.921 |
| | **Ours** w/ VGGT | 21.05 | 0.728 | 0.301 | 0.704 | 0.925 | 22.40 | 0.743 | 0.346 | 0.095 | 0.931 |

large-scale data to refine 3D inconsistencies, enabling robust free-view rendering with floater removal, fewer Gaussians, and faster reconstruction, as illustrated in Fig. 4.

**Depth Evaluation.** We report the depth evaluation results of novel views in Tab. 2. Even without pose or intrinsic as input, our SaLon3R significantly outperforms previous generalizable methods in depth estimation. Fig. 3 shows that our rendered depth achieves smooth and accurate depth results with rich details, eliminating the artifacts arising from overfitting or 3D inconsistencies.

**Zero-shot Evaluation.** To validate the generalization ability, we conduct the zero-shot test on Relica (Straub et al., 2019) and ScanNet++ (Yeshwanth et al., 2023) without further training. The results show that our method outperforms the previous methods in both rendering quality and depth estimation performance, demonstrating our generalization ability to cross datasets.

## 4.3 ABLATION STUDY

**Effect of integrating different backbones.** We conduct an ablation study to investigate the generalization ability of our method to different backbones. Specifically, we take VGGT (Wang et al., 2025a) as the backbone with the fixed pretrained weight, training other modules (Gaussian Head and Refinement network) in the same setup. As shown in Tab. 1, compared to CUT3R, VGGT performs inference over all input images in an offline manner, achieving improved geometry quality, pose accuracy, and efficiency. This leads to consistent performance gains. These results highlight the strong generalization ability of our method, which can be seamlessly integrated with diverse backbones for generalizable GS reconstruction.

**Effect of different components.** We conducted an ablation study to evaluate the contribution of different components (Tab. 5). Starting with CUT3R and a Gaussian head as the baseline, we observed excessive Gaussians and rendering artifacts from retaining all pixel-wise Gaussians. Incorporating the quantization scheme effectively reduces redundancy but introduces quality degradation due to

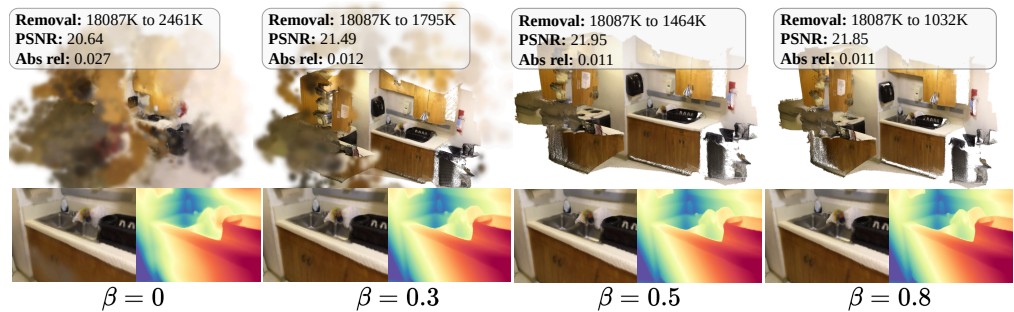

$\beta = 0$        $\beta = 0.3$        $\beta = 0.5$        $\beta = 0.8$

Figure 5: **Effects of Adaptive Gaussian Growing.** Applying $\beta$ from 0 to 0.5 removes most redundant Gaussians and significantly improves the performance. A larger $\beta = 0.8$ could further remove redundant Gaussians with little degradation.

Table 4: **Ablation study on the Number of Context Views.** The context views range from 10 to 200 in the `office02` scene on Replica.

| Views | PSNR↑ | LPIPS↓ | Abs Rel ↓ | FPS ↑ | GS (K)↓ |
|---|---|---|---|---|---|
| 10 | 25.13 | 0.429 | 0.069 | 10.23 | 939 |
| 30 | 24.63 | 0.317 | 0.057 | 9.85 | 2138 |
| 50 | 23.70 | 0.305 | 0.061 | 9.76 | 2905 |
| 100 | 23.05 | 0.321 | 0.069 | 9.72 | 3286 |
| 200 | 23.24 | 0.318 | 0.066 | 9.53 | 3397 |

Table 5: **Ablation Study for the effects of different components** on ScanNet.

| Method | PSNR ↑ | LPIPS ↓ | Abs Rel↓ | GS(K)↓ |
|---|---|---|---|---|
| Baseline | 20.78 | 0.429 | 0.103 | 1966 |
| + Quant. | 16.95 | 0.459 | 0.073 | 616 |
| + Sal. Quant. | 18.06 | 0.436 | 0.065 | 616 |
| + GS Refiner | 21.65 | 0.358 | 0.054 | 616 |
| + Adapt. Grow | 22.45 | 0.347 | 0.078 | 866 |
| + $\mathcal{L}_{smooth}$ (Ours) | 22.54 | 0.337 | 0.030 | 866 |

downsampling. Guided by learned saliency, important regions are preserved and fused, improving rendering quality. Adding a point transformer further refines Gaussian anchors by exploiting structural priors, while our adaptive growing mechanism optimizes Gaussian distribution with structure-aware saliency. Finally, edge-aware regularization enforces local smoothness, yielding additional gains in both rendering and reconstruction quality.

**Effect of view numbers.** We conduct an ablation study to investigate the impact of varying the number of context views on SaLon3R's performance. For a 300-frame scene segment from Replica (Straub et al., 2019), we fixed 1/10 of the frames as target views and uniformly sampled different numbers of context views. As shown in Tab. 4, our method maintains stable processing speed and Gaussian count even with more context views, showing its suitability for long-term online streaming. Increasing context views from 10 to 30 improves depth estimation and LPIPS by leveraging richer input, while PSNR decreases slightly under denser inputs due to photometric inconsistencies but remains stable beyond 50 views.

**Effect of adaptive Gaussian growing.** We conduct an ablation study by varying the opacity threshold $\beta$ for adaptive Gaussian growing to enable the density control. As illustrated in Fig. 5, when increasing $\beta$, it is observed that the background Gaussians are eliminated to decrease, which shows the effectiveness of our saliency-aware mechanism to maintain the region with complex geometry and textures. Therefore, the PSNR and Abs rel are improved by eliminating the noise using $\beta$. However, when setting the threshold to 0.8, there is a little drop in PSNR, indicating that choosing an appropriate $\beta$ could help improve the rendering quality while keeping in small amount of Gaussians.

# 5 CONCLUSION

In this paper, we propose SaLon3R, a novel framework for structure-aware, long-term 3DGS reconstruction. Our method enables efficient and structure-aware scene representation learning in a fully feed-forward manner, without requiring any camera parameters as input. Specifically, we present a saliency-aware quantization scheme for Gaussian primitives using sparse anchors. To further refine the Gaussian representations and mitigate artifacts, we incorporate a lightweight point transformer to facilitate local attention across serialized anchors. The proposed SaLon3R framework supports online incremental reconstruction from streaming image sequences and exhibits superior performance in extensive evaluations. Since our method is designed for static reconstruction, its performance may degrade in dynamic scenes. As future work, we plan to extend it to dynamic, generalizable Gaussian reconstruction for globally consistent 4D reconstruction.

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

# A APPENDIX

## A.1 EXPERIMENTAL ENVIRONMENT

We conduct all the experiments on NVIDIA RTX A100 GPU. The experimental environment is PyTorch 2.5.1 and CUDA 12.6.

## A.2 ADDITIONAL IMPLEMENTATION DETAILS

During training, we fix the weights of the CUT3R model, training the point transformer model and other heads. We set the resolution of the grid sampling to be 0.005 for ScanNet (Dai et al., 2017) and 0.01 for the Replica (Straub et al., 2019) dataset. During training, we set an initial learning rate of $1e^{-5}$ and apply linear warmup with cosine decay. We set the batch size as 1 and the input view number $N = 8$, among which we randomly sample $2 - 6$ views as context views and the rest as target views. During evaluation, we take all the images (target and context) as input to CUT3R, using the estimated poses and intrinsic parameters to render the target view with the Gaussians built from context views.

## A.3 ADDITIONAL EXPERIMENTS

**Effects of Quantization Resolutions.** To investigate the effects of different resolutions, we conduct the ablation study by varying the resolutions from 0 to 0.02, where 0 indicates that no quantization is conducted. As reported in Tab. 6, the reconstruction efficiency shows an increasing trend as the resolution increases. Due to the information loss during the quantization, the performance of novel view synthesis in both RGB and depth will be affected for resolutions over 0.015. Therefore, we choose 0.01 as the resolution for Replica (Straub et al., 2019) Dataset to balance the efficiency and effectiveness.

Table 6: **Ablation study on the Resolution of Quantization.** We control the resolution from 0 (no quantization) to 0.02 from 30 views in the office 02 scene on Replica (Straub et al., 2019).

| Resolution | PSNR↑ | SSIM↑ | LPIPS↓ | Abs Rel ↓ | $\delta_1$ ↑ | FPS ↑ | GS (K)↓ |
|---|---|---|---|---|---|---|---|
| 0 | 24.43 | 0.829 | 0.270 | 0.048 | 0.990 | 7.56 | 5629 |
| 0.005 | 24.48 | 0.832 | 0.264 | 0.049 | 0.990 | 8.95 | 4169 |
| 0.01 | 25.04 | 0.825 | 0.331 | 0.050 | 0.988 | 9.85 | 2005 |
| 0.015 | 24.84 | 0.772 | 0.436 | 0.064 | 0.972 | 10.03 | 942 |
| 0.02 | 24.53 | 0.758 | 0.470 | 0.092 | 0.953 | 10.15 | 506 |

**Visualization of the Predicted saliency.** As shown in Fig. 6, we visualize the refined saliency to investigate the effects. The saliency map is rendered following alpha blending. It is observed that the refined saliency shows higher value and dense distribution for the region with more complex geometry and appearance, *i.e.*, the boundary of objects, regions with texture. In contrast, region with simple texture and flat surfaces shows smaller saliency values and sparse distribution.

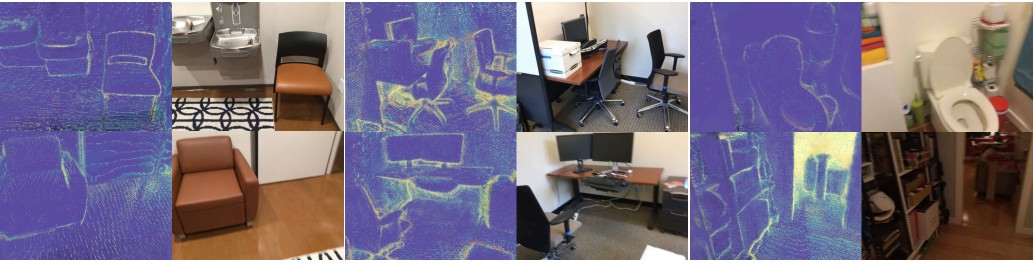

Figure 6: **Visualization of the rendered 3D Saliency.**

**Additional Qualitative Comparison.** We report additional qualitative results of the reconstructed 3DGS and extrapolated view comparison in Fig. 8. It showcases that our method outperforms

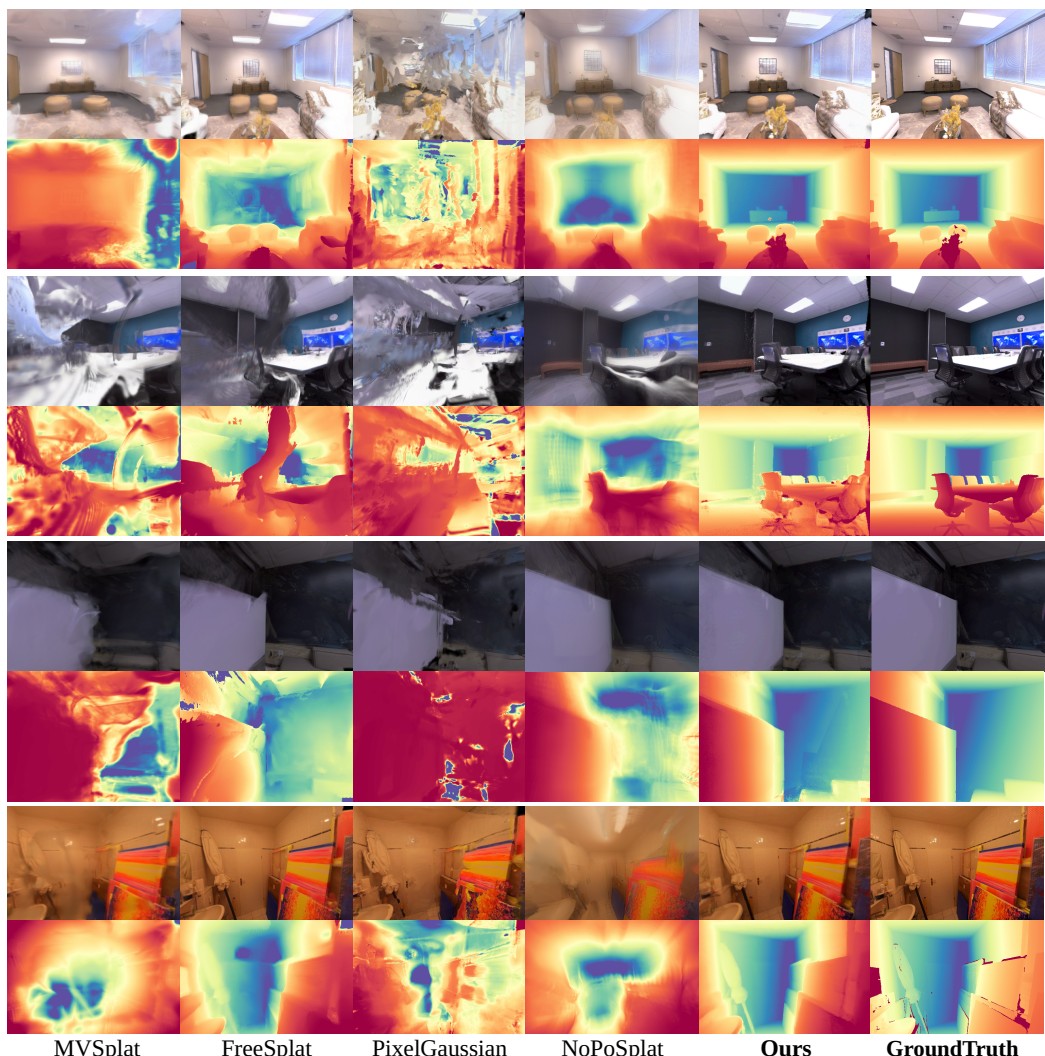

| MVSplat | FreeSplat | PixelGaussian | NoPoSplat | **Ours** | **GroundTruth** |

Figure 7: **Additional Qualitative Results of Novel View Synthesis** with Rendered RGB and Depth.

FreeSplat (Wang et al., 2024c) at the global reconstructed 3DGS and the out-of-distribution views. Although FreeSplat (Wang et al., 2024c) yields good results in view interpolation (see Tab. 1), the poor reconstructed 3D structure causes bad performance in view extrapolation (see Fig. 4 and Fig. 8). In contrast, our structure-aware approach achieves superior 3D reconstruction results, performing well in both interpolation and extrapolation view synthesis. Additional qualitative results of novel view synthesis are shown in Fig. 7, comparing our method with the previous baselines to test the zero-shot performance.

Table 7: **Novel view depth evaluation on ScanNet.** PF: Pose-free methods.

| PF | Method | 30 views | | | | | 50 views | | | | |
|---|---|---|---|---|---|---|---|---|---|---|---|
| | | Abs Rel↓ | Sq Rel↓ | RMSE↓ | RMSE log↓ | $\delta_1$ ↑ | Abs Rel↓ | Sq Rel↓ | RMSE↓ | RMSE log↓ | $\delta_1$ ↑ |
| ✗ | FreeSplat | 0.411 | 0.760 | 0.929 | 0.436 | 0.894 | 0.346 | 0.443 | 0.821 | 0.391 | 0.904 |
| ✓ | NoPoSplat | 1.185 | 0.935 | 1.205 | 0.803 | 0.559 | 1.716 | 4.801 | 1.359 | 1.103 | 0.463 |
| | **Ours** | 0.060 | 0.017 | 0.171 | 0.135 | 0.953 | 0.068 | 0.045 | 0.232 | 0.142 | 0.972 |

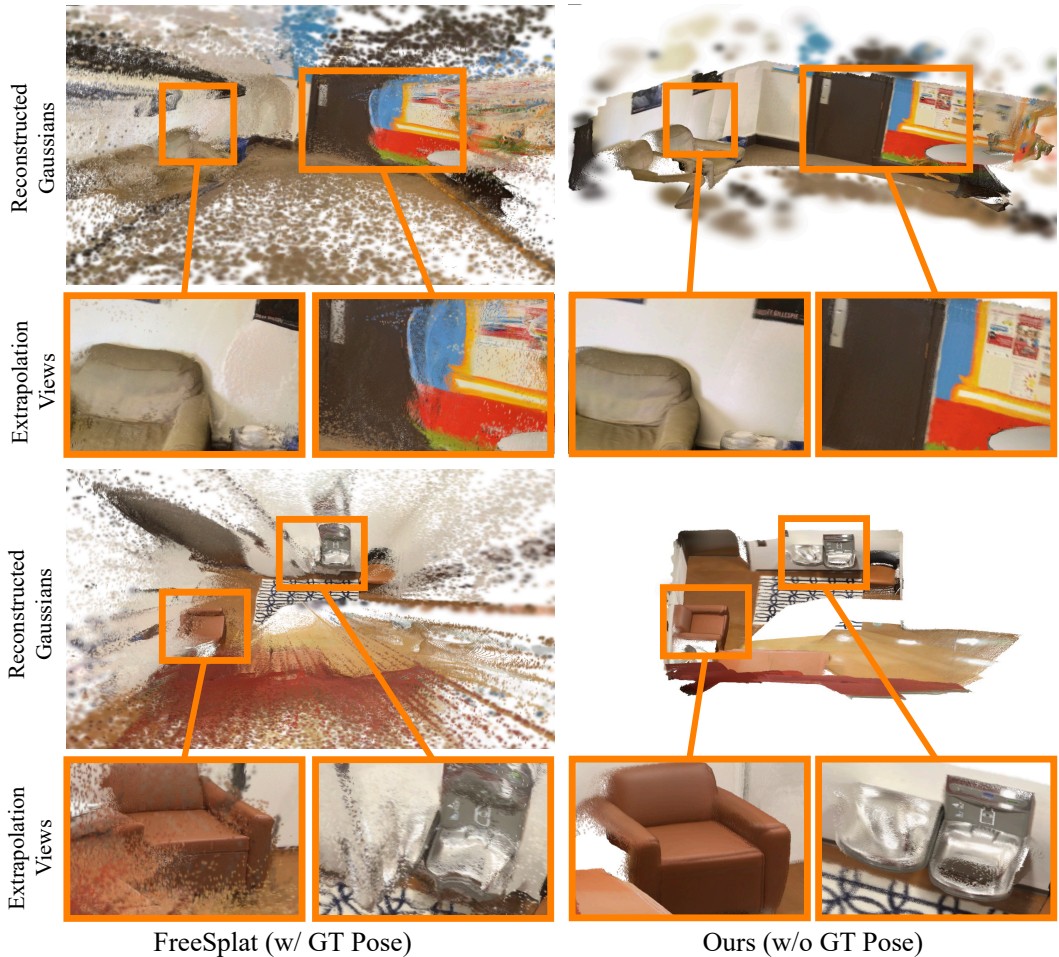

Figure 8: **Additional Visualization of Global Gaussian Splatting and Extrapolation Views.** We zoom in extrapolated views for better comparison with FreeSplat (Wang et al., 2024c) (w/ GT pose).

Table 8: **Novel view depth evaluation on ScanNet.** PF refers to Pose-free methods.

| PF | Method | 10 views | | | | |
|---|---|---|---|---|---|---|
| | | Abs Rel↓ | Sq Rel↓ | RMSE↓ | RMSE log↓ | $\delta_1$ ↑ |
| ✗ | PixelSplat | 0.680 | 1.391 | 0.369 | 0.548 | 0.715 |
| | MVSplat | 0.331 | 0.413 | 0.351 | 0.491 | 0.811 |
| | FreeSplat | 0.102 | 0.276 | 0.256 | 0.165 | 0.973 |
| | PixelGaussian | 0.392 | 0.307 | 0.306 | 0.617 | 0.766 |
| ✓ | NoPoSplat | 0.390 | 0.303 | 0.152 | 0.686 | 0.794 |
| | **Ours** | 0.030 | 0.017 | 0.149 | 0.084 | 0.977 |

**Additional Quantitative Results on Depth Evaluation.** We report the full depth comparison results in Tab. 7, Tab. 8, and Tab. 9. The results show that our method significantly outperforms the previous methods.

A.4 LIMITATIONS AND FUTURE WORKS

Despite the effectiveness of our method, the major limitation exists in the accuracy of the predicted depth and poses from the 3D reconstruction network. Given over long frames as input, due to the lack of global alignment, CUT3R is prone to drift in estimated camera poses and reconstructed pointmaps, leading to view misalignment and floating points. This will inevitably influence the

Table 9: **Novel view depth evaluation on Replica (Straub et al., 2019) and Scannet++ (Yesh-wanth et al., 2023).** PF: Pose-free methods.

| PF | Method | Replica | | | | | Scannet++ | | | | |
|---|---|---|---|---|---|---|---|---|---|---|---|
| | | Abs Rel↓ | Sq Rel↓ | RMSE↓ | RMSE log↓ | $\delta_1$ ↑ | Abs Rel↓ | Sq Rel↓ | RMSE↓ | RMSE log↓ | $\delta_1$ ↑ |
| ✗ | PixelSplat | 4.248 | 2.178 | 0.988 | 0.716 | 0.343 | 0.766 | 9.338 | 2.608 | 0.785 | 0.530 |
| | MVSplat | 2.119 | 1.731 | 0.566 | 0.904 | 0.610 | 0.875 | 2.383 | 1.546 | 1.104 | 0.577 |
| | FreeSplat | 1.485 | 0.470 | 0.383 | 0.373 | 0.790 | 0.350 | 1.400 | 0.954 | 0.360 | 0.828 |
| | PixelGaussian | 2.865 | 1.785 | 0.636 | 1.016 | 0.591 | 1.182 | 1.924 | 1.965 | 1.461 | 0.574 |
| ✓ | NoPoSplat | 1.723 | 0.794 | 0.507 | 0.586 | 0.792 | 0.658 | 0.894 | 1.095 | 0.696 | 0.647 |
| | **Ours** | 0.712 | 0.134 | 0.258 | 0.159 | 0.917 | 0.104 | 0.081 | 0.176 | 0.325 | 0.921 |

rendering performance by introducing artifacts. To address this, we propose processing frames with intervals or employing the keyframe selection to manage large-scale scene reconstruction effectively, ensuring compatibility with the view constraint while maintaining reconstruction quality. Further-more, we could incorporate bundle adjustment to globally align the long sequences. This could effectively help enhance the pose estimation accuracy and 3D consistency across views, making our method robust for complex, multi-room scenarios.

Additionally, our work is designed for static reconstruction and may get degraded in a dynamic scene. Extending our work to the dynamic generalizable Gaussian reconstruction is also an interest-ing future direction. A core challenge lies in enabling the backbone model to track the point-wise 3D motion and to accurately identify the dynamic and static regions. This is essential to help remove the redundancy and artifacts induced by temporally accumulated errors. For achieving globally consistent 4D reconstruction, static regions could be incrementally fused with our saliency-aware quantization and refined over time with the point transformer, while dynamic regions require contin-uous temporal updates from predicted motion. Additionally, modeling temporal residuals in appear-ance will be crucial for capturing photometric changes and ensuring coherent rendering in dynamic scenes.

# B LLM USAGE

We used a large language model (LLM) to assist with writing tasks, specifically to polish language and enhance readability. The authors take full responsibility for the scientific content, research, and analysis, and the LLM was not used for generating scientific ideas or designing experiments.

