# OpenReview forum: "SaLon3R: Structure-aware Long-term Generalizable 3D Reconstruction from Unposed Images"
_ICLR.cc/2026/Conference — ICLR 2026 Conference Withdrawn Submission_

### Official Review · Reviewer_tBnx · 2025-10-29

**Soundness:** 3
**Presentation:** 2
**Contribution:** 3
**Rating:** 4
**Confidence:** 3

**Summary:**

This paper proposes SaLon3R for online generalizable 3D Gaussian Splatting (3DGS) reconstruction from unposed image sequences. The method achieves redundancy removal and performs well on novel view synthesis tasks.

**Strengths:**

1. The paper proposes a novel architecture, SaLon3R, to address the challenging problem of long-term, online, and generalizable 3DGS reconstruction from unposed images.
2. The ablation studies clearly demonstrate the contribution of each individual component.

**Weaknesses:**

1. The method exhibits a strong dependence on the backbone network. The overall performance is highly contingent on the quality of the initial pose and geometry estimates provided by the pretrained reconstruction model, CUT3R.
2. The method is explicitly designed for static scene reconstruction. However, real-world online "long-term" videos often contain dynamic objects, which limits the practical application scope of this work.

**Questions:**

1. Several recent works appear to be missing from the discussion, such as AnySplat[1]. AnySplat utilizes VGGT as a backbone for feed-forward 3DGS, which seems highly relevant given that your architecture also employs VGGT in an ablation study.

2. The datasets used for evaluation are all indoor scenes (e.g., ScanNet, Replica). It would be better if there are more types of scenario evaluations to prove generalization, such as in the wild scene.

[1] Jiang, Lihan, et al. "AnySplat: Feed-forward 3D Gaussian Splatting from Unconstrained Views." arXiv preprint arXiv:2505.23716 (2025).

---

### Official Review · Reviewer_ndDE · 2025-10-31

**Soundness:** 2
**Presentation:** 3
**Contribution:** 2
**Rating:** 2
**Confidence:** 5

**Summary:**

This paper proposes SaLon3R, a feed-forward pose-free 3D Gaussian reconstruction framework designed for long image sequences. The method builds on CUT3R to estimate camera parameters and pointmaps, followed by a Gaussian prediction head, saliency-based voxel quantization, and a lightweight point transformer for refinement. The system targets online Gaussian reconstruction with redundancy reduction, claiming >10 FPS performance and competitive novel-view synthesis and depth estimation on indoor datasets such as ScanNet, ScanNet++, and Replica.

Although the pipeline functions and is evaluated on standard indoor benchmarks, the contribution appears incremental relative to recent pose-free generalizable 3DGS models and CUT3R-based pipelines. Several design elements overlap with contemporary work (e.g., AnySplat), and the method does not demonstrate generalization beyond constrained indoor scenes.

**Strengths:**

1. Reasonable formulation combining CUT3R with Gaussian prediction, voxel quantization, and spatial refinement.

2. Online inference and redundancy pruning are practically valuable for Gaussian-based scene representations.

3. Competitive indoor results on depth and novel-view synthesis benchmarks, with clear comparisons to relevant baselines such as FreeSplat and NoPoSplat.

4. Implementation efficiency with reported >10 FPS online performance and reduced Gaussian counts.

**Weaknesses:**

1. Incremental contribution. The core advances over CUT3R appear modest. The method essentially adds a Gaussian head plus filtering and refinement modules on top of a strong existing reconstruction backbone. The online capability and geometry fidelity are largely inherited from CUT3R, rather than introduced by novel algorithmic components.

2. Limited novelty relative to AnySplat and related work. The approach is conceptually very close to AnySplat, which also leverages feed-forward pose-free reconstruction and voxel-space pruning. The proposed “saliency” score serves a similar role to AnySplat's confidence-based importance weighting for voxel aggregation. The paper does not sufficiently articulate conceptual differences or empirical advantages over this line of work.

3. Restricted experimental scope and lack of generalization evidence. Although CUT3R itself demonstrates strong cross-domain generalization across indoor/outdoor and static/dynamic settings, this work evaluates almost exclusively on indoor scenes (ScanNet/ScanNet++/Replica). It is unclear whether the introduced modules maintain or degrade CUT3R’s broad generalization capacity. No results are presented for outdoor scenes, dynamic scenarios, or more challenging real-world data distributions.

4. Unclear scalability and robustness. While the paper focuses on long sequences, it does not examine long-range drift, global scene consistency, or stability in more complex environments where CUT3R’s estimates deteriorate. The method appears tailored to small-scale indoor reconstructions rather than general long-term scene mapping.

**Questions:**

1. How does the proposed saliency-aware pruning differ fundamentally from AnySplat’s confidence-based pruning? What measurable improvements arise specifically from your design choices?

2. Does your pipeline retain CUT3R’s generalization to outdoor and dynamic scenes? If not, why is performance restricted to indoor static domains?

3. Can you provide ablation or comparative experiments showing performance in more diverse environments, including outdoor datasets and dynamic scenarios?

4. How does your method behave when CUT3R’s pose and depth predictions drift over longer sequences? Is there any mechanism for global alignment or drift correction?

5. Does the Gaussian-based training bring any benefit to pose or pointmap accuracy compared to CUT3R without freezing the corresponding heads?

---

### Official Review · Reviewer_C9v5 · 2025-11-01

**Soundness:** 3
**Presentation:** 2
**Contribution:** 2
**Rating:** 4
**Confidence:** 5

**Summary:**

This paper introduces SaLon3R, an online 3D Gaussian reconstruction model that performs scene reconstruction on-the-fly. The proposed framework is built upon the online point reconstruction model CUT3R, with additional modules for predicting Gaussian latents and a saliency map. To address the issue of excessive Gaussians as the number of input views increases, the method first quantizes Gaussians under the guidance of the saliency map and then refines them using a point transformer. Experimental results demonstrate that the proposed method achieves state-of-the-art performance on online Gaussian reconstruction tasks, evaluated via novel-view image and depth rendering quality.

**Strengths:**

- The proposed method achieves SOTA performance on online Gaussian reconstruction benchmarks.

- The Gaussian refinement module effectively reduces the number of Gaussians while improving reconstruction quality.

**Weaknesses:**

- Re-training baseline methods such as PixelSplat, MVSplat, and NoPoSplat may not yield fair comparisons, as additional in-house noise could be introduced during re-training on new datasets. Moreover, these baselines were trained on a fixed, small number of views, so comparisons on larger view counts may be less meaningful. It would be helpful to also evaluate SaLon3R on the datasets that the baseline models are trained on.

- In the Adaptive Growth stage, what happens if pruning considers only the predicted alpha values? An ablation study is needed to justify the necessity of the proposed Structure-aware Saliency Map, which is claimed as a key contribution.

- Some technical details are missing, such as the view selection strategy used during inference for Gaussian refinement.

- In Fig. 2, the Saliency-aware Quantization module includes an update operation that is not explained in Sec. 3.3, even though the figure refers to that section.

**Questions:**

- Can the proposed method be applied to outdoor scenes, such as DL3DV datasets?
- How does the proposed method compare with multi-view Gaussian reconstruction models such as AnySplat [1]?

[1] Jiang, Lihan, et al. “AnySplat: Feed-forward 3D Gaussian Splatting from Unconstrained Views.”

---

### Note · Authors · 2025-11-18

I have read and agree with the venue's withdrawal policy on behalf of myself and my co-authors.